# Overlapping Crises: Climate Disaster Susceptibility and Incarceration

**DOI:** 10.3390/ijerph19127431

**Published:** 2022-06-17

**Authors:** Kristen N. Cowan, Meghan Peterson, Katherine LeMasters, Lauren Brinkley-Rubinstein

**Affiliations:** 1Department of Epidemiology, Gillings School of Global Public Health, University of North Carolina at Chapel Hill, Chapel Hill, NC 27599, USA; khelen@live.unc.edu; 2Center for Health Equity Research, Department of Social Medicine, School of Medicine, University of North Carolina at Chapel Hill, Chapel Hill, NC 27599, USA; meghan_peterson@med.unc.edu (M.P.); lauren_brinkley@med.unc.edu (L.B.-R.); 3Carolina Population Center, University of North Carolina at Chapel Hill, Chapel Hill, NC 27599, USA

**Keywords:** incarceration, climate, vulnerability, geography, disasters, preparedness

## Abstract

Climate-related disasters are becoming more frequent all over the world; however, there is significant variability in the impact of disasters, including which specific communities are the most vulnerable. The objective of this descriptive study was to examine how climate disaster susceptibility is related to the density of incarceration at the county level in the United States. Percent of the population incarcerated in the 2010 census and the Expected Annual Loss (EAL) from natural hazards were broken into tertiles and mapped bivariately to examine the overlap of areas with high incarceration and susceptibility to climate disasters. Over 13% of counties were in the highest tertile for both incarceration and EAL, with four states containing over 30% of these counties. The density of incarceration and climate disaster susceptibility are overlapping threats that must be addressed concurrently through (1) decarceration, (2) developing standardized guidance on evacuated incarcerated individuals during disasters, and (3) more deeply understanding how the health of everyone in these counties is jeopardized when prisons suffer from climate disasters.

## 1. Introduction

In August 2021, the Intergovernmental Panel on Climate Change (IPCC) released a report that detailed how global surface temperatures have increased more rapidly from 1970 to present than in any other 50-year period over the last 2000 years. This means a higher frequency of extreme events, such as heatwaves, heavy precipitation, droughts, and tropical hurricanes [1]. The burden of extreme weather events will likely not be distributed evenly; there is significant variability in the impact of disasters, including which communities are the most vulnerable [2]. Populations who are more susceptible to the effects of climate disasters include those with lower socioeconomic status, aging adults, and people who are incarcerated [3,4]. Both incarceration and climate change are rooted in historical and current policy choices, resulting in many unmet needs, including climate vulnerability, for incarcerated individuals [5]. Specifically, people who are incarcerated may be more vulnerable to climate hazards due to overcrowding in prisons and limited infrastructure, which leads to basic needs such as air conditioning and medication access being unmet [6]. 

Social and structural factors contribute to harm in the context of climate disasters. For example, neighborhood disadvantage is associated with increased morbidity and mortality after climate disaster events due to disadvantaged neighborhoods receiving a disproportionately small share of disaster relief and mitigation resources [7,8]. Additionally, a multitude of studies have demonstrated that social capital can be one of the biggest predictors of resilience to climate-related hazards [9,10]. Indexes such as the social vulnerability index have been created to identify where socially marginalized groups are located after a disaster [11]. Not only do some communities experience worse outcomes in the aftermath of a disaster due to social vulnerability, some spatial research has even shown that more deprived neighborhoods are more likely to experience climate hazards, such as flooding, due to the sociodemographic makeup of communities that are maintained in more environmentally vulnerable areas [12].

Absent from this literature are studies examining the relationship between climate disaster susceptibility and the criminal legal system, which includes prisons, jails, community supervision (e.g., probation, parole), and policing. The United States (US) has the largest criminal legal system in the world, and prisons are disproportionately located in areas of concentrated rural disadvantage [13]. Furthermore, state governments in the US have historically failed to evacuate incarcerated people in preparation for disasters during previous climate events [14]. Prisons are susceptible to other environmental risks, including historically being placed in hazardous geographies, such as near toxic waste sites, leading to exposure to dangerous environmental hazards that are known to cause adverse symptoms, including gastrointestinal, neurological, and respiratory problems [14,15]. While some research has explored environmental hazards located near prisons, little research has been done to investigate how mass incarceration will interact with the effects of climate change. Additionally, no research has identified how areas with high rates of incarceration may be increasingly vulnerable to disasters. In order to answer these questions, the objective of this study was to examine how climate disaster susceptibility is related to the density of incarceration at the county level. This descriptive study seeks to identify areas of the United States where higher proportions of people who are incarcerated are at higher risk of experiencing climate disasters in order to inform immediate policy changes to protect those who are incarcerated.

## 2. Methods

### 2.1. Data Sources

Climate disaster susceptibility was measured with the Federal Emergency Management Agency (FEMA) Expected Annual Loss (EAL) scale, which measures the amount of financial loss due to 18 different natural hazards for all counties of the United States [16]. This scale was used to evaluate climate disaster susceptibility because it is available for all counties and captures the impacts of many types of climate disasters, which we defined as any type of disaster that occurs as a result of a climate hazard. The 18 natural hazards captured in the EAL are avalanche, coastal flooding, cold wave, drought, earthquake, hail, heat wave, hurricane, ice storm, landslide, lightning, riverine flooding, strong wind, tornado, tsunami, volcanic activity, wildfire, and winter weather. EAL is calculated using three main components: exposure, annualized frequency, and historic loss ratio. Exposure refers to the value of buildings, population, or agriculture land that is potentially exposed to natural hazards. Annualized frequency refers to the expected frequency or probability of a natural hazard occurring per year. Historic loss ratio refers to the percentage of the exposed property value expected to be lost due to a natural hazard if one were to occur. Together, these variables create an EAL estimate that shows the average economic loss resulting from natural hazards each year, with a score ranging from 0–100 [16]. For this analysis, we used EAL data at the county level. Our rationale in using county-level measures was that these are politically meaningful units on which interventions can be targeted. Data were obtained from the Federal Emergency Management Agency (FEMA).

Data on incarceration were obtained from the Marshall Project, a nonprofit organization that examines the U.S. criminal legal system [17], which provides the number of people who are incarcerated in a county at the 2010 census and the total population for that county [18]. In these data, the number of people who are incarcerated include people in federal detention centers, federal and state prisons, local jails, and carceral residential facilities. The percentage of the 2010 population that was incarcerated was calculated and used as our measure of incarceration level by county.

### 2.2. Spatial Methods

We first assessed spatial clustering using Moran’s I to determine if the distribution of incarceration is randomly distributed in the United States or not [19]. In other words, are places of high incarceration and loss located near each other? The proportion of the population that is incarcerated and the FEMA EAL score were aggregated at the county level. Data on percent of population incarcerated and EAL were mapped at the county level in tertiles of continuous values to descriptively examine which counties have the highest proportion of incarcerated people and are at highest risk of experiencing financial loss from a natural hazard. FEMA EAL and percent of population incarcerated were broken into tertiles and mapped bivariately to examine locations that have a high level of both incarceration and natural hazard loss risk. ArcGIS Pro 2.8.3 [20] was used for analyses.

## 3. Results

Examining clustering of incarceration, we found that counties with high incarceration rates are clustered near each other (z-score = 1.58) The median county-wide percent incarcerated rate was 0.26%, with a range from 0–45.6%. EAL Score is a continuous measure from 0–100; the median county-level EAL score was 11.52, with a range from 0.02–93.64. Using the FEMA qualitative rating categories, 0.92% of counties had very high, 4.36% relatively high, 14.54% relatively moderate, 40.10% relatively low, and 10.07% very low EAL scores.

EAL scores and county-level incarceration rate were categorized into tertiles and mapped to demonstrate which counties are at highest risk for its incarcerated population (Figure 1). The highest proportion of counties (15.82%) were in the lowest tertile for both incarceration and EAL. An additional 14.10% of counties were in the medium tertile for incarceration but the highest tertile for EAL. A total of 13.14% of counties were in the highest tertile for both incarceration and EAL score. The lowest number of counties, 6.08%, were in the lowest level for incarceration and the highest EAL group. Examining trends by state, of counties in the highest level for both incarceration and EAL score, most were in Texas (11.38%), followed by Florida (7.51%), North Carolina (7.02%), and California (6.54%).

## 4. Discussion

This study is the first that looks at the relationship between climate disaster susceptibility and density of incarceration. Specifically, we found that density of incarceration and climate disaster susceptibility are not randomly dispersed. They are overlapping threats that must be addressed concurrently. We found that four states—Texas, Florida, North Carolina, and California—contain over 30% of the counties with the highest EAL scores and incarceration rates. While these states are at higher risk of climate hazards such as flooding and wildfires due to their geography, it is important to note that these are also places with the highest incarceration rates complementing a high risk of climate hazards. By identifying areas where both of these threats exist, we are able to target public health interventions for immediate change in areas at highest risk for people who are incarcerated. A limitation of this research is that it is descriptive and cannot be used to draw formal conclusions on the relationship between climate-hazards and incarceration. While no research exists specifically on this topic, our findings corroborate existing research indicating that the burden of climate disaster will not be distributed evenly [2]. The overlapping nature of climate disaster susceptibility and incarceration requires the following from the public health community:

(1) States must prioritize counties with high incarceration rates and high susceptibility to climate hazards for immediate intervention. One important strategy is to reduce the prison population as much as possible and close prison facilities in these communities [21]. Past events have demonstrated that incarcerated individuals are likely to be detained in inhabitable conditions during extreme weather events [22]. It is expected that events will increase in magnitude and severity due to climate change, which will affect the health of people in prisons and jails, particularly in areas sustaining repeated extreme events. Decarceration can reduce the number of individuals who are susceptible to these natural disasters. Incarceration is known to have harmful impacts on health [23]. Therefore, decarceration is a measure that should be taken to improve public health, and decarceration will decrease morbidity and mortality among people who are incarcerated from disasters specifically.

(2) In addition to decarceration, policymakers should provide explicit directives on evacuation and aid to people who are incarcerated during disasters. The US lacks uniform policy on protecting people who are incarcerated and people who work in prisons during emergencies [24], and there are no federal guidelines for temperature regulation in heat waves [5]. Currently, a large amount of the preparedness planning that is done by the Departments of Corrections primarily includes language on protection of property and labor that can be contributed by incarcerated people in the event of a disaster. State and community level preparedness planning should include plans for all hazards that they are at risk of experiencing in their geographical location as well as language specifically to prepare for protection and evacuation of vulnerable populations, including incarcerated people.

A set of policies should be developed with input from people who have been incarcerated and must be standardized across states with more stringent policies in areas facing the highest EAL. Funding should also be dedicated to not only writing preparedness plans, but conducting drills and exercises to prepare for these events as they become more common due to climate change. Further, public health agencies outside of the criminal legal system, such as Departments of Health, should be granted powers to independently monitor carceral facilities for climate disaster preparedness and respond to disasters as they occur [25].

(3) Policymakers must reinvest in health and social services in areas with high rates of incarceration. Historically, communities where policymakers have heavily invested in incarceration have also been communities with disinvestment in terms of health and social services [21]. The criminal legal system affects entire communities, not only people who are incarcerated [23]. In places with high incarceration rates, it is unknown what the collateral consequences of these disasters will be for communities. For example, people who are incarcerated during a disaster may require increased emergency medical services and inpatient hospital care given the health risks inside these facilities and the lack of evacuation orders, which can further strain health systems in the community in times of crisis [26,27]. Thus, entire communities with high incarceration and high EAL may be at a significant disadvantage in recovery from disasters.

As described previously, community-level social capital is one of the biggest indicators of resilience to climate disasters [9,10]. Access to healthcare and social services before disasters occur could equip communities with the tools they need to be more resilient to disasters, and people living within prisons are no different in regard to these needs. Therefore, counties with high incarceration and climate susceptibility must be prioritized for targeted disaster mitigation and relief programming to ensure that health services are adequately prepared for climate disasters.

(4) Additional research is needed on the impacts of climate hazards on this marginalized population. The descriptive results from this study indicate that higher levels of climate vulnerability and higher rates of incarceration are clustered in similar areas; however, further research on what this means for these communities and people who are incarcerated would be helpful for informing policy changes. Further research is needed that examines how decarceration, preparedness planning, and improvement of social services could reduce poor health outcomes among incarcerated people in the event of a climate-related disaster.

## 5. Public Health Implications

We recommend that state health departments and policymakers urgently act to mitigate the harms of climate change and to protect people who are incarcerated and their communities as the threat of climate disasters escalate. When public health crises impact people who are incarcerated, it possibly strains the health and well-being of the entire community. Prisons are not separate from communities, and people who are incarcerated have been overlooked by public health and not considered relevant to climate disaster risk. Overall, measures that can be taken to reduce the number of people who are incarcerated, improve emergency preparedness planning for prisons, improve healthcare access to those who are incarcerated, and further investigate the compounding hazards of incarceration and climate disasters need to be taken immediately to reduce the harmful impacts of climate vulnerability on communities overall.

## Figures and Tables

**Figure 1 ijerph-19-07431-f001:**
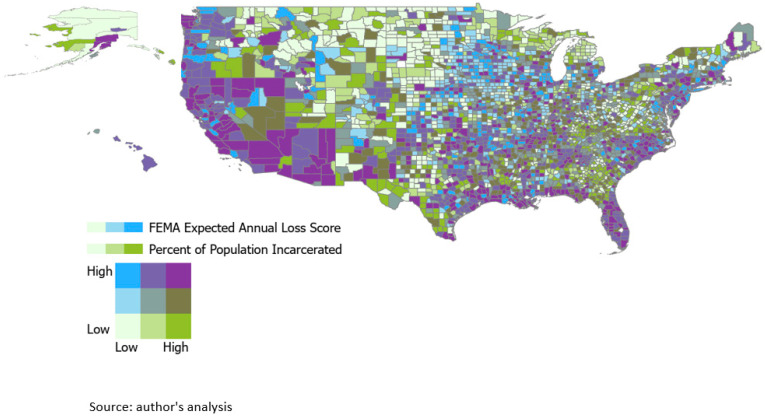
Expected annual loss score tertiles by percentage of population incarcerated tertiles.

## Data Availability

The data presented in this study are openly available through FEMA and the Marshall Project at https://hazards.fema.gov/nri/data-resources and https://observablehq.com/@themarshallproject/adults-in-correctional-facilities-from-decennial-census (accessed on 6 June 2022).

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
