# Peer review of "Overlapping Crises: Climate Disaster Susceptibility and Incarceration"

_ijerph, 2022, doi:10.3390/ijerph19127431_

Round 1

Reviewer 1 Report

Overall, this is a very interesting paper, and it touches on a topic that not many people care about. There are some minor grammatical errors that the authors can work on. More comments are as follows:

Introduction: Interestingly, you mentioned about vulnerable group. It would be even stronger if you could briefly explain why incarcerated people are susceptible to climate change, especially how society views them put minimal incentive for the government to reduce their vulnerabilities.

Line 66: There are many definitions of exposure. It might be beneficial to explain why you decide to use this one.

Line 76: What is Marshall Project? You might want to elaborate.

Line 117-119: Are there any potential explanations for why these states have high EAL scores.

Author Response

Thank you for your comments, my replies have been addressed in the attached document. 

Reviewer 2 Report

This study examines how climate disaster susceptibility is related to the density of incarceration at the county level in the US. Given the paucity of literature on the subject, this research is significant. However, the authors need to strengthen the background of why this study needs to be done. The justification given by the author has not been able to explain the connection between climate-related disasters and people who are incarcerated.

In Line 28-30, the author only mentions that the effect of disasters (in general, not specific to climate disasters) affects people who are incarcerated.

On the other hand, in Line 3-42, the author describes how the impact of climate disasters on the community (in general, not specific to people who are incarcerated).

In Lines 50-56, the explanation given by the authors addresses the environmental hazards.

The author must first describe the scope of the climate-related disaster discussed in this study.

In the method section, the data used was obtained in 2010. It means that the data is almost 12 years old. Is there any particular reason why this data is used? Is there no updated data?

Author Response

Thank you for your comments, my reply is in the attached document. 

Reviewer 3 Report

When research method is considered vis-a-vis the purpose of research, this article seems to have logical validity and originality in the research methodological approach and the process of deriving analytical results. However, the implications in terms of the policy are somewhat general lacking specificity, especially when presenting the improvement tasks derived from the research results by way of listing. It would be more appropriate to present specific suggestions in the conclusion based on the research results. 

Additional comments:
Heading/subheading: Discussion parts to be seperated and author can also highlight the research gaps/constratins and limitation instead of seperate implication sub-heading and also point out how the study addresses the  crisis and how it considers mitigation measures. Create subheadings based on the theoretical framework and research findings. They give the reader clear ideas to understand the highlights of the study.

The most important thing is the complement of the conclusion. The meaning of the conclusion, the policy meaning, the theoretical meaning, and the limitations of this study should be more abundant. Previous studies should also be investigated more abundantly.

Author Response

(The authors gave the same response as above.)

Round 2

Reviewer 3 Report

authors have improved the work.

This manuscript is a resubmission of an earlier submission. The following is a list of the peer review reports and author responses from that submission.